# Benchmarking Differential Abundance Tests for 16S microbiome sequencing data using simulated data based on experimental templates

Eva Kohnert, Clemens Kreutz*

Institute of Medical Biometry and Statistics, Faculty of Medicine and Medical Center, University of Freiburg, Germany

* clemens.kreutz@uniklinik-freiburg.de

## Abstract

Differential abundance (DA) analysis of metagenomic microbiome data is essential for understanding microbial community dynamics across various environments and hosts. Identifying microorganisms that differ significantly in abundance between conditions (e.g., health vs. disease) is crucial for insights into environmental adaptations, disease development, and host health. However, the statistical interpretation of microbiome data is challenged by inherent sparsity and compositional nature, necessitating tailored DA methods. This benchmarking study aims to simulate synthetic 16S microbiome data using metaSPARSim (Patuzzi I, Baruzzo G, Losasso C, Ricci A, Di Camillo B. MetaSPARSim: a 16S rRNA gene sequencing count data simulator. BMC Bioinformatics. 2019;20:416. https://doi.org/10.1186/s12859-019-2882-6 PMID: 31757204) MIDASim (He M, Zhao N, Satten GA. MIDASim: a fast and simple simulator for realistic microbiome data. Available from: https://doi.org/10.1101/2023.03.23.533996), and sparseDOSSA2 (Ma S, Ren B, Mallick H, Moon YS, Schwager E, Maharjan S, et al. A statistical model for describing and simulating microbial community profiles. PLOS Comput Biol. 2021;17(9):e1008913. https://doi.org/10.1371/journal.pcbi.1008913 PMID: 34516542) , leveraging 38 real-world experimental templates (S3 Table) previously utilized in a benchmark study comparing DA tools. These datasets, drawn from diverse environments such as human gut, soil, and marine habitats, serve as the foundation for our simulation efforts. We employ the same 14 DA tests that were previously used with the same experimental data in benchmark studies alongside 8 DA tests that were developed subsequently. Initially, we will generate synthetic data closely mirroring the experimental datasets, incorporating a known truth to cover a broad range of real-world data characteristics. This approach allows us to assess the ability of DA methods to recover known true differential abundances. We will further simulate datasets by altering sparsity, effect size, and sample size, thus creating a comprehensive collection for applying the 22 DA tests. The outcomes, focusing on sensitivities and

**Data availability statement:** All relevant data from this study will be made available upon study completion.

**Funding:** The author(s) received no specific funding for this work.

**Competing interests:** The authors have declared that no competing interests exist.

specificities, will provide insights into the performance of DA tests and their dependencies on sparsity, effect size, and sample size. Additionally, we will calculate data characteristics (S1 and S2 Table) for each simulated dataset and use a multiple regression to identify informative data characteristics influencing test performance. Our prior study, where we used simulated data without incorporating a known truth, demonstrated the feasibility of using synthetic data to validate experimental findings. This current study aims to enhance our understanding by systematically evaluating the impact of known truth incorporation on DA test performance, thereby providing further information for the selection and application of DA methods in microbiome research.

## Introduction

### Study rational

Microbial communities play a crucial role in various ecosystems, including human health, agriculture, and environmental processes. In medical research, investigating the microbiome has become pivotal due to its proven impact on the development of multiple diseases, and its potential as prognostic and predictive factor.

The major step in microbiome analysis is quantification of the abundance of microbial taxa such as bacterial species and identification of significant changes. A large set of statistical methods have been suggested for differential abundance (DA) analysis and it has been shown that the outcomes strongly depend on the chosen method. However, method selection guidelines are still missing and existing benchmark studies only show a fragmentary and inconsistent picture about the performance of these DA methods.

### Background

This work builds on the seminal study by Nearing et al. [1], which systematically compared the performance of 14 differential abundance (DA) tests applied to 38 experimental 16S microbiome datasets. These datasets, sourced from diverse environments such as the human gut, soil, wastewater, freshwater, plastisphere, marine, and built environments, were used in a two-group design to identify variations in species abundances. While Nearing et al.'s study [1] provided a comprehensive comparison of the outcomes and agreements of DA tests, it did not address the correctness of these tools due to the absence of a known ground truth in the data.

In our previous validation study, "Computational Study Protocol: Leveraging Synthetic Data to Validate a Benchmark Study for Differential Abundance Tests for 16S Microbiome Sequencing Data" [2], we explored the potential of using two simulation tools for microbial count data, metaSPARSim [3] and sparseDOSSA2 [4] to replicate real experimental data and validate findings obtained by experimental data. Our previous study referred to Nearing et al.'s [1] benchmarking efforts and demonstrated that the two proposed simulation tools tend to underestimate the prevalence of zero counts, but showed good agreement with experimental data, successfully reproducing global

tendencies of the statistical tests when sparsity is adjusted by adding an appropriate proportion of zeros. For brief illustration, we summarize our preliminary findings of that study in Fig 1. For each template, 10 simulated data sets were generated, and 46 distinct data characteristics [2] were calculated for each dataset. Fig 1A displays the PCA plot of the scaled data characteristics for metaSPARSim. Templates are represented as squares, with the 10 corresponding simulated datasets shown as dots of the same colour. At this summary level of all data characteristics, the synthetic datasets generated by metaSPARSim are generally very close to their respective templates (Fig 1A). Fig 1B shows a closer examination for four example data characteristics. The left sections of these boxplots show the difference of the data characteristics between an experimental template and other templates, serving as a measure of the variability if datasets from different projects are compared. The middle sections of the plots visualize deviations of the data characteristic if a template is compared with the 10 simulated datasets, serving as a measure of how precisely simulated data reflect characteristics of the respective template. The right section of the plot displays the variation within the simulated data sets generated for each template, serving as a measure for variability of the data characteristic introduced by simulation noise. The left panel of Fig 1B demonstrates that metaSPARSim tends to underestimate the proportion of zeros which can be corrected by adding zeros as shown on the right. In general, simulated data tend to overestimate the bimodality of sample correlations, a metric used to measure taxa-specific effect sizes. This bimodality characteristics exhibited the greatest discrepancy between real and simulated data. Other characteristics such as the 95% quantile or the Inverse Simpson diversity of the samples are very similar when comparing sparsity-adjusted simulated data with the respective template (middle sections in all boxplots).

Building on these findings, our current study aims to enhance this framework by incorporating a known ground truth. For this purpose, we will generate synthetic data that spans a broad range of effect sizes, sparsity levels, and sample sizes. This approach will enable a more detailed evaluation of the sensitivity and specificity of the 22 DA tests, providing deeper insights into their performance across various data characteristics.

### Objectives, research questions and hypotheses

**Primary objectives, research questions and hypotheses. Aim 1**: Assess the performance of 22 differential abundance tests on simulated data, corresponding to 38 experimental 16S microbiome sequencing data templates.

**Research question:** Which differential abundance testing methods among the selected 22 methods provide the most accurate results in terms of sensitivity and specificity for a prespecified 5% threshold for the false discovery rate?

**Hypothesis**: Certain differential abundance tests outperform others in terms of sensitivity and specificity.

**Secondary objectives, research questions and hypotheses. Aim 2**: Assess how the performance of 22 differential abundance tests depend on sparsity of the dataset, effect size and sample size

**Research question**: How do changes in dataset sparsity, effect size, and sample size impact the sensitivity and specificity of differential abundance tests for a given 5% threshold for the false discovery rate?

**Hypothesis**: Performance advantages of differential abundance tests significantly depend on the sparsity of the data, effect size and sample size.

**Exploratory objectives and research questions. Aim 3**: Identification of data characteristics that are predictive for performance advantages of specific differential abundance tests.

**Research question:** Which data characteristics are predictive for performance advantages and can be used to predict methods with beneficial performance?

**Hypothesis**: There are data characteristics which can be used to predict performance advantages.

## Methods: datasets

### Population

**Aim 1 (Primary).** Based on 38 experimental 16S microbiome sequencing data templates, simulated datasets are generated. The experimental data templates originate from diverse environmental contexts such as the human gut, soil,

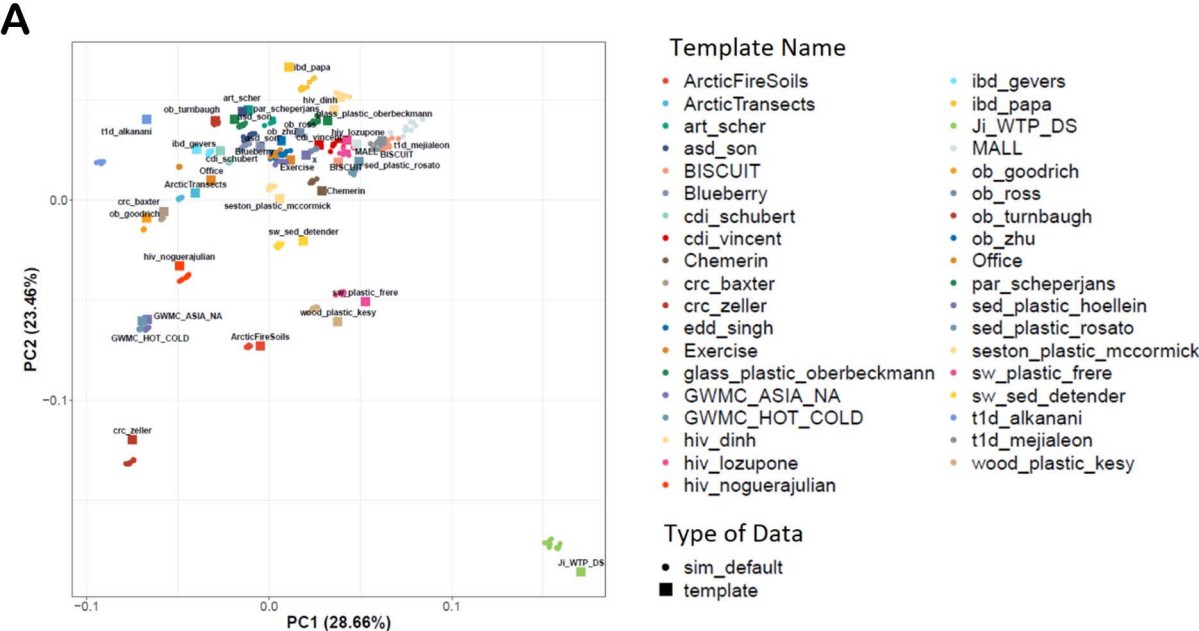

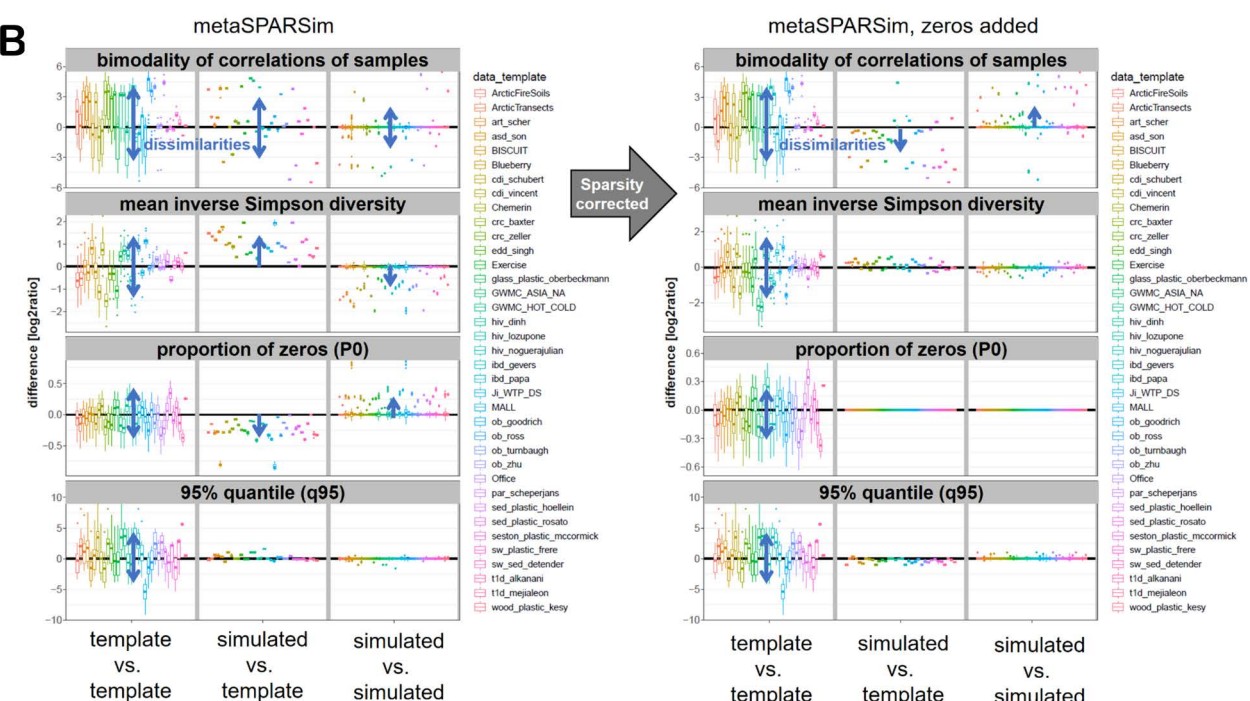

**Fig 1. Preliminary results assessing the similarity of simulated data and corresponding templates. A.** Overall similarity of simulated data and templates for metaSPARSim. PCA plot on 46 scaled data characteristics for 38 templates and 10 corresponding simulations. Templates are plotted as squares and simulations as dots in the same colour. B. Accuracy of four representative single data characteristics. Overall magnitudes of visible bias and heterogeneities are highlighted by blue arrows. The left sections in all panels show the natural variability of a specific data characteristic among the templates. Here, the log2-ratios of the data characteristics from one template to all other is summarized as boxplot. In the middle the precision of the data characteristic in the simulations compared to the corresponding template is displayed. The right sections show log2-ratios of the data characteristic between all simulations belonging to the same template.

wastewater, freshwater, plastisphere, marine and built environments. These templates were selected because they were previously used as benchmark datasets in Nearing et al. for complementary benchmarking analyses. Moreover, we used these datasets for simulating data to validate the results of Nearing et al. [1] and for studying the capabilities of simulated data, to reflect characteristics of their data templates realistically. These 38 experimental data templates exhibit a broad spectrum of data characteristics, including varying sample sizes ranging from 24 to 2296 and feature counts from 327 to 59,736. A detailed overview of dataset characteristics for these templates is provided in Supplement 2.

Three simulation tools, metaSPARSim [3], sparseDOSSA2 [4] and MIDASim [5], are employed to generate for each data template 10 simulated dataset, closely mimicking the original characteristics. This is achieved by calibrating the simulation parameters for each data template individually. If a tool underestimates the proportion of zeros, an appropriate proportion of zeros will be added as suggested in [2]. Then, the two simulation tools that best reflect experimental data templates are chosen for subsequent analyses. The quantitative number to judge at this point is the total number of rejected equivalence tests over all datasets and data characteristics [2].

A known ground truth in the simulated data with regulated as well as not regulated taxa is integrated as described in the following. The simulation parameters for both tools are calibrated three times: 1) To obtain parameters that correspond to no difference between the two groups of samples, calibration is done using experimental data for all samples jointly, i.e., independent on the group information. 2)+3) To obtain parameters, that are different in the two groups, calibration is done in each group separately, i.e., the simulation tools are calibrated to the samples belonging to the first group, and calibrated to the samples belonging to the second group.

To obtain both, differentially abundant taxa and taxa with the same expectation in both groups, these calibrated simulation parameters 1) and 2)+3) are merged by estimating the proportion of differentially abundant taxa from the distribution of p-values, and then randomly draw differential abundant taxa. In more detail, the following procedure will be applied:

1. All DA methods are applied to the experimental data templates for calculating p-values

2. To obtain one p-value for each taxon and treat all DA methods equally, a p-value of a randomly selected DA method is assigned

3. The proportion pi0 of not differentially abundant features is estimated by the pi0est function in the qvalue R-package [6]. This function estimates the proportion of true null hypotheses, i.e., those following a uniform distribution [7].

4. Simulation parameters are calibrated (a) for all samples and (b) for each group separately

5. For the proportion pi0, simulation parameters are chosen from calibration 1), and for the proportion (1-pi0), simulation parameters are taken from calibration 2)+3). Differentially abundant features with simulation parameter from 2)+3) are drawn via *isDiffAbundant = runif(n = length(pvalues)) p.adjust(pvalues, method = "BH")* to ensure that taxa with smaller p-values are more likely assigned to be differentially abundant.

Therefore, the population for aim1 is composed of $10$ (*simulations*) x $38$ (*templates*) x $2$ (*simulators*) $= 760$ simulated 16S microbiome sequencing data, for which the number of truly differentially abundant features is known.

**Aim 2 (Secondary).** Here, additional variations for each of the $2$ (*simulators*) x $38$ (*templates*) $= 76$ simulations are generated by modifying sparsity, effect size, and sample size, such that each ranges from low over medium to high. Therefore, the population for aim 2 is composed of $76$ x $3$ (*sparsity*) x $3$ (*effect size*) x $3$ (*sample size*) $= 2.052$ simulated 16S microbiome sequencing data settings, for which the number of truly differentially abundant features is known.

For a realistic range of sparsity, effect size and sample size in the simulated data, the minimum, median and maximum value for each of these properties are taken from the 38 experimental data templates and used as lower and upper bound in the simulated data. As an example, the minimal sample size is $N_{min} = 24$ (Human - IBD), the median is $N_{med} = 76.5$ and the maximal sample size is $N_{max} = 2296$ (Freshwater - Arctic). Thus, we simulate data for these three sample sizes.

The effect size is quantified using PERMANOVA R-square [8] as implemented in the adonis2 function of the vegan R-package [9]. First, the minimal and maximal R-squared values are calculated over all templates. In order to adjust the effect size to minimal, median and maximal values calculated from all templates, first the relationship between a fold-change multiplication factor F and PERMANOVA R-square [8] is evaluated. For this purpose, for each data template one simulated dataset is generated with $F = 0, 0.1, 0.2, \ldots, 1$, followed by the calculation of the respective PERMANOVA R-square [8] value. The appropriate factor F for each template and each desired R-square value is chosen by linear interpolation of these 11 points. If this procedure yields multiple solutions because the relationship may not be monotone, the average of the resulting Fs is used.

For metaSPARSim [3], these fold-parameters are the ratios between the two estimated intensity parameters in both groups for each taxon. For sparseDOSSA2 [4], the fold-parameters are the ratios between the two parameters mu in both groups for each taxon.

**Aim 3 (Exploratory).** For aim 3 the same datasets as for aim 2 are used

## Sources

Nearing et al. [1] collected and harmonized a set of 38 public datasets. We use these datasets published at https://figshare.com/articles/dataset/16S_rRNA_Microbiome_Datasets/14531724 as input to simulate synthetic data for three published simulation tools: metaSPARSim [3], MIDASim [5] and sparseDOSSA2 [4].

## Sample size considerations

We chose the same study population, i.e., the same number of experimental data templates as in previous benchmark studies (e.g., Nearing et al. [1]) to enable comparability.

For aim 1, a strict sample size calculation is not feasible since we consider multiple comparisons between the 22 DA methods and we do not know the magnitude of differences of the target variable (pAUC) and their variabilities over multiple simulate datasets.

Based on our experience about runtimes of the simulation and the 22 DA methods we chose $N = 10$ simulated datasets for each template (i.e., 380 simulated dataset in total for each simulation tool) for addressing our primary aim. For this sample size, a relative difference of at least 1.6 is required in a worst-case scenario with intra-class correlation coefficient $ICC \approx 1$ to obtain >90% power when the analysis is performed with the mixed effects model as described below (two-sided analysis with $\alpha = 0.05$). In the case of a vanishing $ICC \approx 0$, a relative effect size 0.236 can be detected with a power of 90%.

For aim 2, we also have to keep runtimes manageable. Moreover, sample size calculations for forward selection and using rankings of the 22 DA methods as response variable would require many assumptions and are therefore not feasible. We therefore chose the study design and sample size based on qualitative and practical aspects. Since evaluating an additional data characteristic (in terms of samples size, effect size and sparsity) is more informative for uncovering general trends than replicating the same combination multiple times, we plan to only simulate one dataset per combination, but explore all 27 combinations for each data template. Statistical interpretation and conclusions are drawn at least on the repeated outcomes (i.e., rankings of the 22 DA methods) over the 38 data templates.

Aim 3 is investigated using the same data and sample size as aim 2 for comparability reasons. Since the analysis of aim 3 relies on forward selection, a reliable statement concerning the statistical power is not feasible from our point of view.

## Eligibility criteria

We will include the same experimental datasets as in Nearing et al. [1].

Since we analyze the performance of DA methods using simulated data, we only want to include simulated data that is realistic, i.e., that has data characteristics similar to real experimental data used as templates. We therefore apply

a similar exclusion strategy for data templates that cannot be realistically resembled by metaSPARSim [3], sparse-DOSSA2 [4], or MIDASim [5] as suggested in [2]. Our overall strategy for including and excluding datasets is summarized in Fig 2.

## Selection process and its results

Datasets used as templates to simulate data are the same as in Nearing et al. [1].

The exclusion strategy for simulated data is based on equivalence tests comparing simulated data with its experimental template and has been presented in [2]. Equivalence tests are conducted for 46 different data properties. Following this strategy, the simulated data for an experimental template is excluded when the number of non-equivalent data characteristics is an outlier when comparing all experimental templates. Fig 3 illustrates such outliers.

**Visualization of the dataset selection process and its results in the form of a flowchart.**

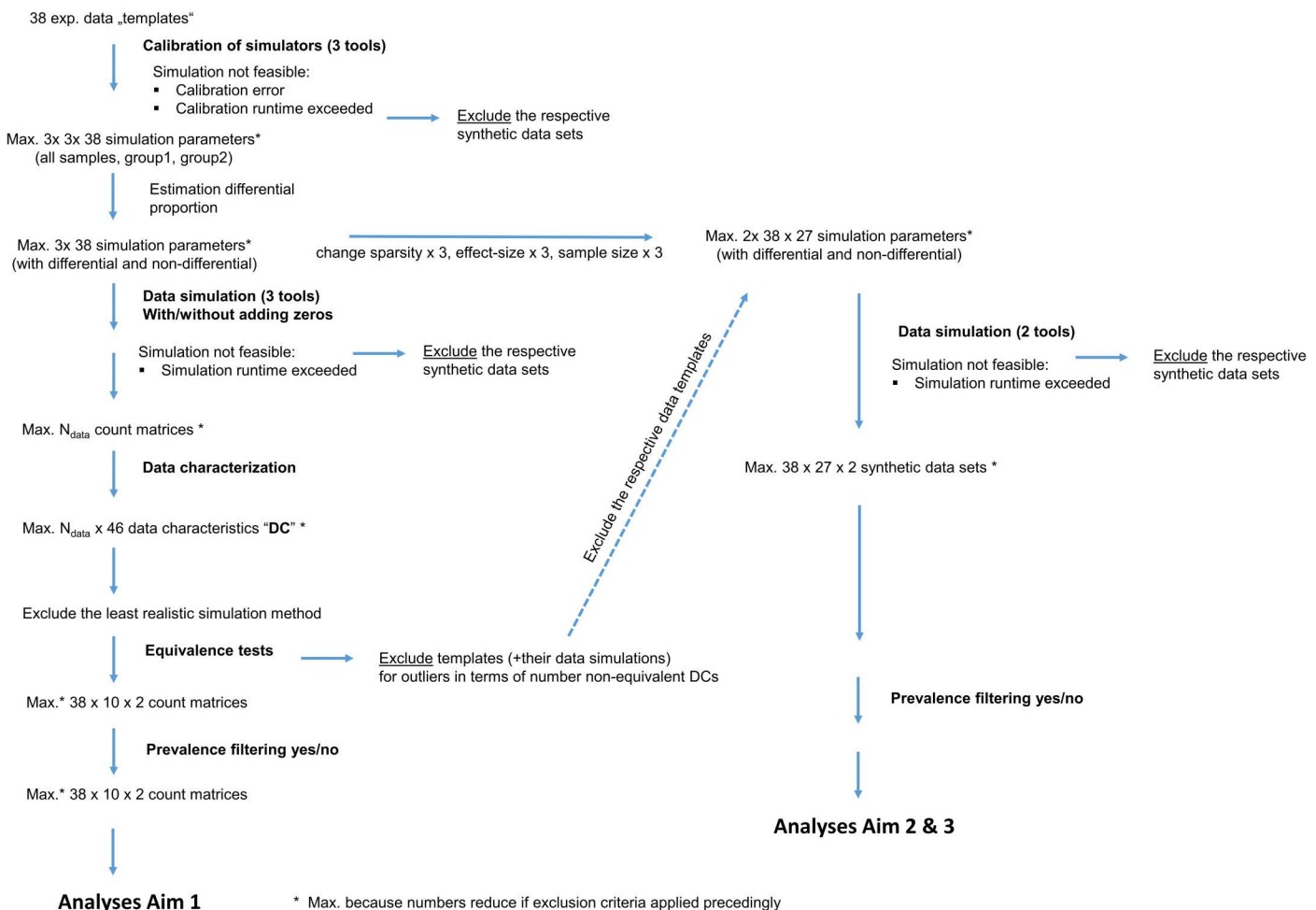

**Fig 2. Overview about the data generating mechanism including the dataset selection process.** Flowchart summarizing the data generating and selection mechanism throughout the entire workflow of the study.

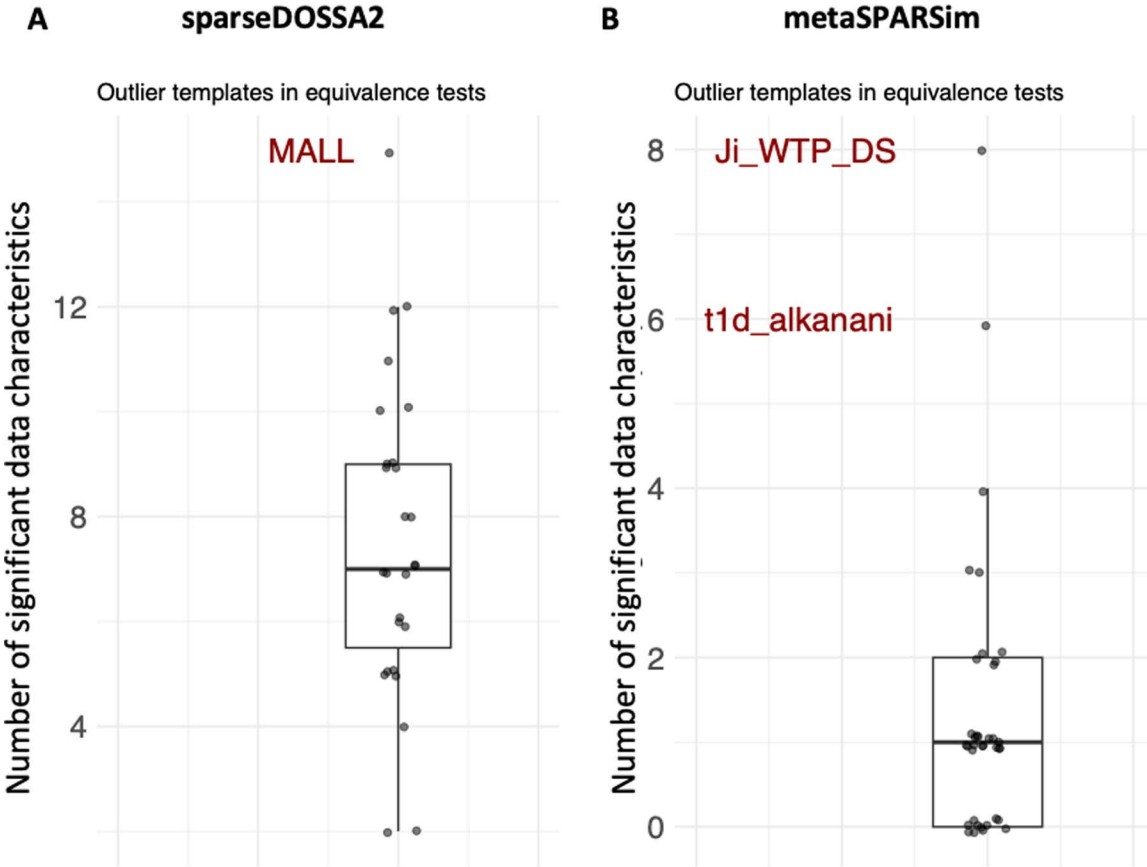

**Fig 3. Illustration of detection of outlier data sets after simulation.** Each dot represents the number of non-equivalent data characteristics for a data template. If this number is an outlier in the boxplot the synthetic data from this template will be removed from the analysis. A If sparseDOSSA2 would result in such an outcome, the synthetic dataset for the template MALL would be removed. B If metaSPARSim would result in such a boxplot, based on the outlier criteria two data templates would be removed from the analysis (Ji_WTP_DS and t1d_alkanani).

## Methods: Benchmark experiment plan

### Benchmark/study design

**General benchmark setup/design.** Our study is an exploratory benchmark study based on simulated data. The overall workflow is summarized in Fig 4. Since the taxa that were simulated as differential abundant are known, we can calculate the sensitivity and specificity for each significance level α. We use the partial area under the receiver operator characteristic curve (pAUC) and calculate ranks(pAUC) over all 22 DA methods to compare their performance.

**Studied methods and collected measures (incl. rationale).** The studied methods are the same 22 differential abundance (DA) tests that were used in Nearing et al. [1], but as implemented in the latest R version: ADAPT [10], ALDEx2 [11], ANCOM-BC2 [12], corncob [13], DESeq2 [14], distinctTest [15], edgeR [16], fastANCOM [17], glmmTMB [18], LEfSe [19], limma voom (TMM) [20], limma voom (TMMwsp) [20], linDA [21], MaAsLin2 [22], MaAsLin2 (rare) [22], MaAsLin3 [23], metagenomeSeq [24], t-test (rare), Wilcoxon test (CLR), Wilcoxon test (rare), ZicoSeq [25], and ZINQ [26]. Significant features will be determined using a 0.05 threshold for adjusted p-values (Benjamini-Hochberg correction).

The collected performance measures are sensitivity and specificity for each differential abundance test.
**Employed validation procedure/technique.** N/A.

## Preprocessing procedure

As in Nearing et al. [1] we analyze unfiltered data (all taxa) as well as filtered data using prevalence-filtered data with a 10% threshold (only taxa with at least 10% counts > 0).

Samples without meta information and samples that are not assigned to one of the two compared groups will be omitted. Moreover, taxa without count > 0 and samples with sum(counts) < 100 will be eliminated to remove "empty" samples

## Analysis workflow summary

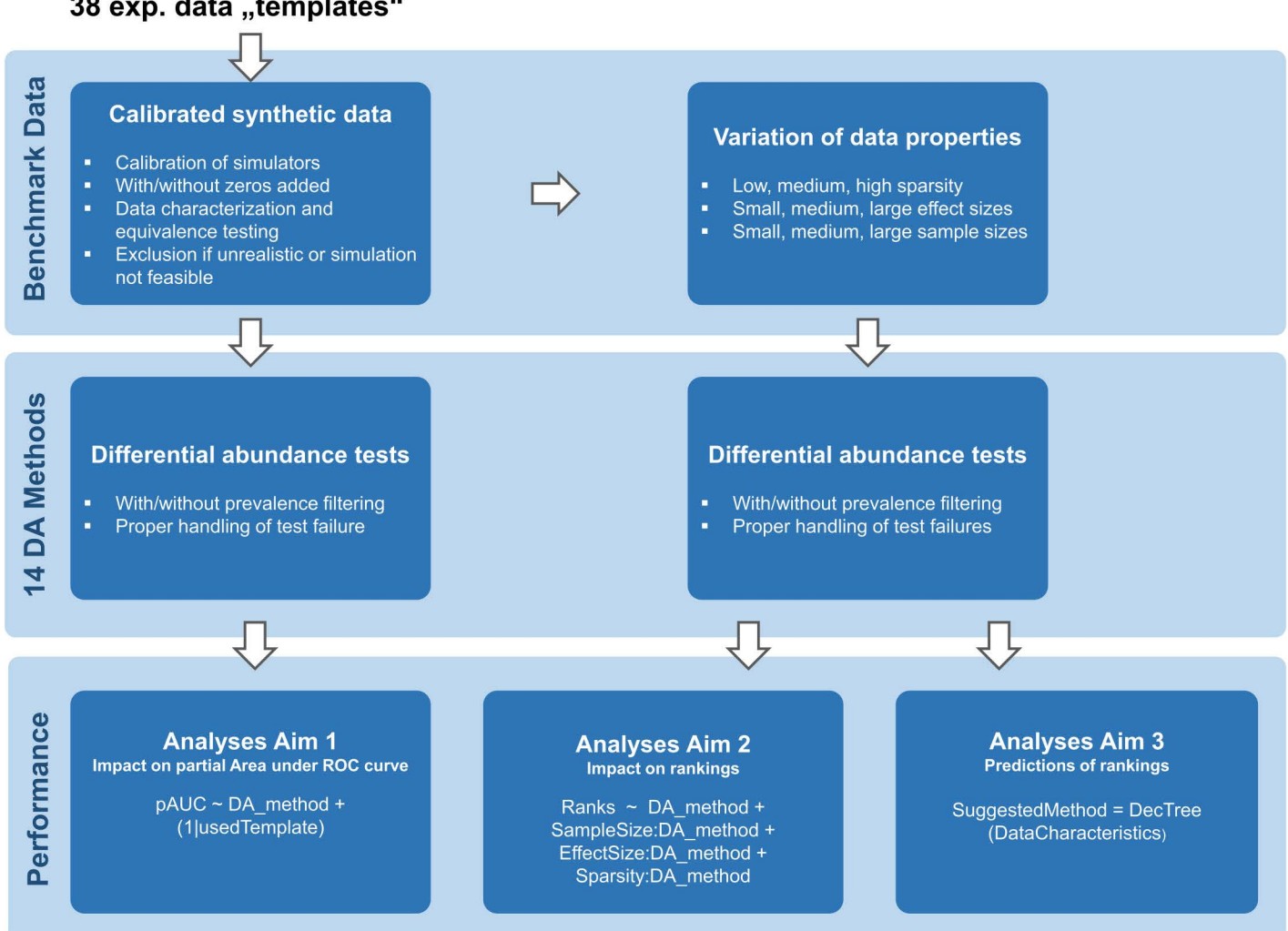

**Fig 4. Overview of the complete analysis workflow including the data simulation process.** Fig 4 provides an overview about the analyses conducted within our study that are described in the following sections.

and taxa. Since both criteria depend on each other, this check is applied repeatedly until no "empty" taxa or samples remains.

Since metaSPARSim [3] requires raw count data as well as normalized data, we use Geometric Mean of Pairwise Ratios (GMPR) normalization as suggested in the metaSPARSim documentation and as implemented in https://github.com/jchen1981/GMPR/blob/master/GMPR.R (version with commit ID 29bfb73) and as previously applied in [2] using the following options: $gmpr.size.factors = GMPR(counts, ct.min = 1, trace = TRUE, nsamples = Inf, nfeature = 20000)$. If these size factors cannot be calculated for a subset of samples due to limited overlap of prevalent taxa, we replace these missing values by the median over the other samples.

Three DA methods apply a rarefying step. This is conducted using the rrarefy() function from the vegan package [9] directly before the DA method is called. As in Nearing et al. [1], only samples with more than 2000 counts are considered and the minimum of those samples is used as subsample size for rarefying.

## Methods configurations

**Parameters and/or configurations for each method.** We implemented the 22 DA methods and tested their applicability. The following code summarized the configurations that will be used in the study:

**ADAPT (Analysis of Microbiome Differential Abundance by Pooling Tobit Models):**

```
res <- NULL
phylo <- bs_data2phyloseq(counts, meta, buildTree = F)
try({
 res <- ADAPT::adapt(input_data = phylo,
  cond.var = "condition",     # Grouping variable
  prev.filter = 0,      # No extra removing of low-prevalence taxa
  depth.filter = 100    # Remove low-depth samples
 )
})
if (is.null(res)) {
 warning("Error: ADAPT did not succeed.")
} else {
 pvalues <- res@details$pval
 logFoldChange <- res@details$log10foldchange * log2(10) # transform to log2
}
```

**ANCOM-BC (Analysis of Compositions of Microbiomes with Bias Correction):**

```
phylo = creadPhyloseqObject(counts, meta, buildTree = F)
Result = ancombc2(data = phylo,
        fix_formula = groupVariable,
        p_adj_method = "BH",
        tax_level = "Species")
```

**ALDEx (ANOVA-Like Differential Expression tool for high throughput sequencing data):**

```
Result = ALDEx2::aldex(reads = counts,
        conditions = as.character(meta[, groupVariable]), mc.samples = 128,
        test = "t",
        effect = TRUE,
        include.sample.summary = FALSE, denom = "all")
```

**Corncob (Count Regression for Correlated Observations):**

```
Result = corncob::differentialTest(formula = design,
        phi.formula = design,
        phi.formula_null = design,
        formula_null = ~ 1,
        test = "Wald",
        data = as.data.frame(counts),
        sample_data = meta,
        boot = F,
        fdr_cutoff = Inf)$p # no cutoff (output of all is needed)
```

**distinctTest (differential analyses via hierarchical permutation tests) REF:**

```
meta <- data.frame(meta, cluster_id = rep("A", nrow(meta)), sample_id = row.names(meta))
# Create SummarizedExperiment
se <- SummarizedExperiment::SummarizedExperiment(
    assays = list(logcounts = log2(counts + 1)),      # Log2 - transformed counts
    colData = meta)
old_seed <- .Random.seed # remember old seed
set.seed(61217) # seed from distinct::distinct_test docu
res <- distinct::distinct_test(x = se, design = design,
    name_assays_expression = "logcounts", min_non_zero_cells = min_non_zero_cells)
.Random.seed <- old_seed # set seed to old value
pvalue <- res$p_val
if (sum(res$filtered) > 0)
    p[res$filtered] <- NA # for fitered taxa: overwrite 1 by NA
```

**EdgeR:**

```
dgeList = DGEList(counts = counts, group = as.numeric(as.factor(meta[, groupVariable])))
dgeList = edgeR::calcNormFactors(dgeList)
dgeList = edgeR::estimateDisp(dgeList)
Result = edgeR::exactTest(dgeList)
```

### fastANCOM:

```
group <- as.factor(meta[,"condition"])
Result <- fastANCOM::fastANCOM(Y = t(counts), x = group)$results
pvalue <- fit$results$final$log2FC.pval
```

### glmmTMB:

```
Result <- list()
for(i in 1:nrow(counts)){
  df <- data.frame(meta, counts = counts[i,], libSize = colSums(counts))
  fit <- glmmTMB::glmmTMB(formula = counts ~ condition,
        data = df, ziformula = as.formula("~1"),
        family = "nbinom2")
  coefs <- as.data.frame(coefficients(summary(fit))$condition)
  if(fit$fit$convergence != 0) # check convergence
    coefs[,1:4] <- NA # overwrite if not converged
  icoef <- grep("condition", rownames(coefs))
  Result[["pvalue"]][i] <- coefs[icoef, "Estimate"]
}
```

### Limma voom TMM:

```
DGE_LIST = DGEList(counts = counts, group = as.numeric(as.factor(meta[,groupVariable])))
Upper_Quartile_norm_test = edgeR::calcNormFactors(DGE_LIST, method = "upperquartile")
summary_upper_quartile = median(Upper_Quartile_norm_test$samples$norm.factors) # CK
if(is.na(summary_upper_quartile) | is.infinite(summary_upper_quartile)){
  Ref_col = which.max(colSums(sqrt(ASV_table)))
  DGE_LIST_Norm = edgeR::calcNormFactors(DGE_LIST, method = "TMM", refColumn = Ref_col)
}else{
  DGE_LIST_Norm = edgeR::calcNormFactors(DGE_LIST, method = "TMM")
}
mm = model.matrix(~comparison, groupings)
voomResult = voom(DGE_LIST_Norm, mm, plot = F)
fit = eBayes(lmFit(voomResult, mm))
Result = topTable(fit, coef = 2, n = nrow(DGE_LIST_Norm), sort.by = "none")
```

**Limma voom TMM:**

$DGE\_LIST = DGEList(counts = counts, group = as.numeric(as.factor(meta[, groupVariable])))$

$Upper\_Quartile\_norm\_test = edgeR::calcNormFactors(DGE\_LIST, method = "upperquartile")$

$summary\_upper\_quartile = median(Upper\_Quartile\_norm\_test\$samples\$norm.factors)$

$if(is.na(summary\_upper\_quartile) | is.infinite(summary\_upper\_quartile))\{$

$\quad Ref\_col = which.max(colSums(sqrt(ASV\_table)))$

$\quad DGE\_LIST\_Norm = edgeR::calcNormFactors(DGE\_LIST,$

$method = "TMM", refColumn = Ref\_col)$

$\} \, else \, \{$

$\quad DGE\_LIST\_Norm = edgeR::calcNormFactors(DGE\_LIST, method = "TMM")$

$\}$

$mm = model.matrix(as.formula(paste0("\sim", groupVariable)), meta)$

$voomResult = voom(DGE\_LIST\_Norm, mm, plot = F)$

$fit = eBayes(lmFit(voomResult, mm))$

$Result = topTable(fit, coef = 2, n = nrow(DGE\_LIST\_Norm), sort.by = "none")$

**LEfSe (Linear discriminant analysis Effect Size):**

$phylo = makePhyloseqObject(counts, meta, buildTree = F)$

$Delta = 0 \, \# \, relax \, thresholds, i.e. less \, output \, p - values \, if \, lefse \, does \, not \, run$

$res = NULL$

$while(is.null(res) \, \&\& \, Delta < 1)\{$

$\quad \# \, lda_{cutoff} = 0 \, to \, output \, as \, many \, features \, as \, possible, 0 \, might \, not \, work$

$\quad try(res = microbiomeMarker::run\_lefse(phylo, group = groupVariable,$

$\qquad norm = "TSS", \quad kw\_cutoff = 1 - Delta,$

$\qquad\qquad wilcoxon\_cutoff = 1 - Delta, lda\_cutoff = Delta))$

$\quad Delta = Delta + 0.05 \, \# \, This \, iterative \, increase \, is \, only \, done, if \, LEfSE \, fails \, in > 10\%$

$\}$

$if(is.null(res))\{$

$\quad warning("Error: lefse \, did \, not \, succeed.")$

$\} \, else \, \{$

$\quad marker = list(pvalue = array(dim = nrow(counts)))$

$\quad marker = microbiomeMarker::marker\_table(res)$

$\quad marker = marker[match(paste("s\_", row.names(counts), sep = ""), marker\$feature),]$

$\}$

$pvals = marker\$pvalue$

**linDA:**

$Result <- LinDA::linda($
 $otu.tab = counts,$
 $meta = meta,$
 $formula = '\sim condition',$
 $alpha = 1, \# Significance\ threshold\ for\ FDR\ control$
 $prev.cut = 0, \# Prevalence\ filter$
 $lib.cut = 100, \# Minimum\ library\ size\ filter$
 $winsor = TRUE\ \ \# Winsorization\ to\ reduce\ outlier\ effects$
 $)$
$pvalue = Result\$output\$condition\$`pvalue`$

**MaAsLin2 (Microbiome Multivariable Association with Linear Models):**

$Result = Maaslin2(counts, groupings, output = "tmpFolder",$
 $transform = "AST",$
 $fixed\_effects = groupVariable,$
 $min\_prevalence = 0,$
 $standardize = FALSE,$
 $plot\_heatmap = F,$
 $plot\_scatter = F,$
 $max\_significance = Inf)$

**MaAsLin2 rare:**

$counts\_rarefied = doRarefy(counts, groupings)\$counts$
$Result = Maaslin2(counts\_rarefied, groupings, output = "tmpFolder",$
 $transform = "AST",$
 $fixed\_effects = groupVariable,$
 $min\_prevalence = 0,$
 $standardize = FALSE,$
 $plot\_heatmap = F,$
 $plot\_scatter = F,$
 $max\_significance = Inf)$

**MaAsLin3:**

$Result = maaslin3::maaslin3($
 $counts, meta, output, transform = "LOG",$
 $fixed\_effects = "condition", min\_prevalence = 0,$
 $standardize = FALSE, max\_significance = Inf)$

**metagenomeSeq:**

$feature\_data = AnnotatedDataFrame(data.frame("ASV" = rownames(counts),$

$\qquad\qquad\qquad\qquad\qquad "ASV2" = rownames(counts)))$

$rownames(feature\_data) = feature\_data@data\$ASV$

$test\_obj = metagenomeSeq::newMRexperiment(counts,$

$\qquad\qquad phenoData = AnnotatedDataFrame(groupings),$

$\qquad\qquad featureData = featureData)$

$pvals = metagenomeSeq::cumNormStat(test\_obj, pFlag = T)$

**T test:**

$pvals = apply(counts, 1,$

$function(x)\, t.test(x \sim groupings[, groupVariable], exact = F)\$p.value)$

**T test rare:**

$counts\_rarefied = doRarefy(counts, groupings)\$counts$

$pvals = apply(counts\_rarefied, 1,$

$\qquad\qquad function(x)\, t.test(x \sim groupings[, groupVariable], exact = F)\$p.value)$

**Wilcoxon rare:**

$counts\_rarefied = doRarefy(counts, groupings)\$counts$

$pvals = apply(counts\_rarefied, 1,$

$\qquad function(x)\, wilcox.test(x \sim groupings[, groupVariable], exact = F)\$p.value)$

**Wilcoxon CLR:**

$CLR\_table = data.frame(apply(counts + 1, 2, function(x)\{log(x) - mean(log(x), na.rm = T)\}))$

$pvals = apply(CLR\_table, 1,$

$function(x)\, wilcox.test(x \sim groupings[, groupVariable], exact = F)\$p.value)$

**ZicoSeq:**

$group <- as.factor(meta[,"condition"])$

$retained <- array(T, dim = nrow(counts))$

$for(grouplevel\ in\ levels(group))\{$

$\quad nonzero\_counts <- rowSums(counts[, group == grouplevel] > 0)$

$\quad retained <- retained\ \&\ nonzero\_counts >= min(sum(group == grouplevel), 5)$

$\}$

$filtered\_counts <- counts[retained,, drop = FALSE]$

$Result <- GUniFrac::ZicoSeq($

$\quad meta.dat = meta,$

$\quad feature.dat = filtered\_counts,$

$\quad grp.name = "condition",$

$\quad adj.name = NULL,$

$\quad feature.dat.type = "count",$

$)$

$pvalue <- rep(NA, nrow(counts))$

$pvalue[retained\_indices] <- Result\$p.raw$

**ZINQ:**

```
results_list <- lapply(1: nrow(counts), function(i) {
  dat <- data.frame(y = counts[i, ], condition = group)
  tryCatch({
    res <- ZINQ::ZINQ_tests(
      formula.logistic = y ~ condition,
      formula.quantile = y ~ condition,
      C = "condition",
      data = dat
    )
    return(res$pvalue.logistic)
  }, error = function(e) {
    return(NA)
  })
})
pvalue <- as.vector(do.call(rbind, results_list))
```

To guarantee that FDR calculations coincide for all DA methods, we perform FDR adjustments manually for each DA method and each analyzed test via p.adjust(pvalues, method="BH").

Altogether, we use the same configuration parameters (such as the threshold 100 for minimal library size of a sample) as chosen in Nearing et al. [1] to ensure comparability of the outcomes with the following main exceptions:

- Instead of conducting rarefication as a first step, we will apply the rarefy step directly before the DA method is called to be able to calibrate the simulation tools before

- Instead of using a custom script for ANCOM-II, we will use the ANCOM-BC2 R package that was published after the study by Nearing et al. [1].

- For LEfSe, we use p-values instead of the score. By default, LEfSe outputs only p-values for taxa, that are not filtered out according to three cutoff configuration parameters (kw_cutoff, lda_cutoff, wilcoxon_cutoff) used for filtering. Unfortunately, relaxing those cutoffs to values where all p-values are returned can lead to failure of LEfSe in rare cases. In [2] we chose a strategy where those cutoffs are iteratively relaxed, i.e., we iteratively changed them towards default values if LEfSe fails with relaxed cutoffs.. To better reflect real-world usage in this study, we decided to stick to single value for those cutoffs, if failure of LEfSe occurs in <=10% of the analyzed datasets. In case failures occurs in > 10%, we will apply the iterative changes as in [2]. In any case, we will calculate FDRs from the (possibly filtered) p-values, independently on violation of mathematical assumptions since this would presumably also be done in real-world applications. A more detailed justification of this procedure can be found online in our responses to reviewer comments. Taxa that do not fulfill all cutoffs will have NA as p-value.

**Description of hyperparameter tuning, if applicable.** N/A.

## Methods: Analysis plan

**Operationalization of hypothesis and evaluation metric.** We measure the performance of the DA methods by evaluating sensitivity and specificity. The relationship between sensitivity and specificity is evaluated by calculating

the area under the receiver operator characteristic (ROC) curve. Since in practice differentially abundant taxa are only identified using a significance threshold <= 0.05, we only compute and compare the partial pAUC for 1-specificity = [0, 0.05] for each test and dataset.

**Statistical techniques to evaluate hypothesis. Aim 1 (Primary):**

To estimate the average performance difference over all analyzed datasets, a mixed effects model is applied:

$$pAUC \sim DA\_method + (1|usedTemplate) \text{ × } 2013; \text{ } 1$$

Now, the best performing method is used as intercept in order to estimate test performance loss of the other methods using the following mixed effects model:

$$pAUC \sim DA\_method + (1|usedTemplate)$$

As the above models assume normality of the residuals, this is tested by employing the Shapiro-Wilk test. In case of significance ($p < 0.01$), a Box-Cox transformation is applied to pAUC to resemble normality. Subsequently, p-values of the mixed effects model with transformed pAUC values will be calculated.

**Additional descriptive analyses**: A summary of performance results for each statistical test will be visualized using boxplots.

**Aim 2 (Secondary):**

For each simulation setting (in terms of sparsity, effect size and sample size), the 22 DA methods will be ranked according to their pAUC. The ranks are then analyzed using stepwise forward selection with as null model and as maximal model. Starting from the null model, this forward selection approach iteratively adds variables from the full model, evaluating their contribution to explaining pAUC until the optimal subset of predictors is determined.

$$Ranks \sim DA\_method$$

$$Ranks \sim DA\_method + SampleSize : DA\_method + EffectSize : DA\_method + Sparsity : DA\_method$$

Note, that the chosen rank transformation leads to ranks 1, 2, …, 22 for each combination of used data template, sparsity, effect size, and group size. Therefore, neither the used data template is required as predictor, nor the main effects for GroupSize, EffectSize and Sparsity. In case a test fails, NA will be assigned to the respective DA method and the remaining ranks will be scaled to the range [21,27] to not bias the rankings of the other DA methods.

Additionally, descriptive analyses will be employed to explore the defined research aims. For example:

**PCA Plot of data characteristics**: Principal Component Analysis (PCA) plots will be generated to visualize the overall variation in dataset characteristics using the prcomp function from the standard stats R package. If missing values occur, they are imputed for PCA by medians. Each data point (representing a dataset) can be colored based on its corresponding AUC value, providing insights into how dataset features relate to test performance.

**Connected Dotplot**: A connected dotplot can be constructed where each dot represents the performance (e.g., pAUC) of a statistical test on a specific dataset. Dots from the same test are connected, allowing for the visualization of trends or consistency in performance across datasets.

**N-way ANOVA**: N-way ANOVA will be applied for the selected multiple regression model to investigate whether performance benefits of specific tests significantly depend on multiple factor levels of the three simulation parameters. This statistical analysis will help identify significant differences in performance and potential interactions between tests and dataset characteristics.

**Inference criteria.** We conduct the statistical analysis with linear mixed-effects models using the lme4 and lmerTest R packages. Specifically, coefficients and standard errors will be estimated via restricted maximum likelihood (REML), p-values using ANOVA and forward selection based on the Akaike Information Criterion (AIC). All p-values below a significance level $a = 0.05$ will be considered as significant.

## Contingencies and backup plans

### Modification of differential abundance (DA) tests. Inflated runtime

Datasets with a large number of features could lead to inflating runtimes for some statistical tests. We therefore define and apply a runtime threshold. This threshold will be set as the largest possible number, so that the predicted runtime of all DA tests does not exceed 3 months on a compute server with AMD EPYC 9454 48-Core Processor. Since we have to apply all DA methods to the experimental template once to estimate the number of not differentially abundant taxa (pi0), we use the observed runtimes for defining the runtime threshold.

If the runtime threshold for an individual test is exceeded for a specific dataset, we split the dataset into two subsets, each containing half of the taxa and try to complete the calculation within the maximal allowed runtime. To preserve the data structure as much as possible and avoid introducing additional randomness, we divide the data into even and odd rows. Specifically, the first subset contains taxa from rows *1, 3, 5,..., $N_{taxa} - 1$*, while the second subset includes taxa from rows *2, 4, 6,..., $N_{taxa}$*. The DA method is applied separately to each subset, and the resulting p-values from both splits are subsequently merged.

If the runtime threshold is still exceeded for either subset, this split-and-merge procedure is repeated once more for each subset, effectively dividing the original dataset into a maximum of four subsets. For technical reasons, the runtime threshold is evaluated individually for each split and not to the total runtime summed across all subsets.

Here, we define the runtime threshold to be max. 1 hour per test. Then, in a worst-case scenario, the 22 tests for the 10+27 datasets for each of the 38 template (19684 combinations) and each simulation tool would need 820 days on a single core. Since we can conduct the tests on up to 96 cores, such a worst-case scenario would still be manageable.

### Test failure

If a DA test throws an error, we apply the following procedure:

We first investigate whether failure of specific tests is related to performance, e.g., because it might happen that failures are more frequent for datasets which are more difficult to analyze, for example due to increased sparsity. Then, treating pAUC as NA would lead to bias. In order to investigate and prevent this, we apply logistic regression using the following model: failure ~ DA_method + DA_method:pAUC. Since pAUC is missing if a DA method failed, we have to impute those pAUCs. For this purpose, we use the median pAUC if the pAUC is available for respective DA_methods for datasets with the same simulation parameters. Otherwise, a mixed effects model pAUC ~ DA_method + (1 | dataTemplate) for all available pAUCs with the DA_method as fixed effect and the data template as a random effect is used for imputation.

If there are DA methods without significant interactions DA_method:pAUC, we omit the respective outcome and treat and reported them as NA (not available) as it would occur in practice without manual tuning the DA method. For significant interactions, we conduct a second analysis where NAs of those methods are replaced by the worst pAUC. This means that we treat failure of a DA like having the worst performance of all methods und we report this as outcome of or study. In addition, we compare these results with using NAs and report the respective results additionally.

## Alternative analysis strategies and sensitivity analyses

For informative failures of DA methods, we evaluated the sensitivity of the outcomes on how those NAs are treated, as described in checklist item 20.

The dependency of the performance of the DA methods on dataset characteristics is investigated as exploratory analyses of Aim 3 (see checklist item 22).

## Exploratory analyses

For our exploratory Aim 3, we calculate 46 data characteristics for each dataset, as detailed in Supplement 1. These characteristics will be scaled (mean = 0, SD = 1) to ensure comparability. The multivariate regression approach applied for Aim 2 will then be employed to explore potential predictive relationships between these data characteristics and the performance of each statistical test.

To identify the most influential data characteristics, a forward selection algorithm will be utilized to identify a minimal set of characteristics that are predictive for the performances We will also explore the possibility to find decisions rules. Such rules would look like: "choose DA method A if specific data characteristics are met, and method B otherwise". For this purpose, we determine decision trees using the minimal set of predictive characteristics predChar1, predChar2, predChar3, … as predictors and the best performing statistical method as categorical response. This will be conducted as done in [28] using the Recursive Partitioning trees as indicated by the following pseudo-code and results in a hierarchy of conditions "if predCharX> threshold" indicating the optimal test for given characteristics of a dataset.

$$tree\_model \; < - \; rpart(bestTest \; \sim \; predChar1 + \; predChar\,2 \; + \; predChar3 + , data \; = \; df, \; method \; = \; "class")$$
$$\#\,'class'\; specifies \; classification$$

## Methods: Software, hardware and reproducibility

### Software

**List of software, central packages and dependencies (with version numbers) and their purpose.** R packages for data simulation: metaSPARSim_1.1.2, qvalue_2.32.0, SparseDOSSA2_0.99.2, vegan_2.6-4

R packages for DA tests: ADAPT_0.99.1, ALDEx2_1.32.0, ANCOMBC_2.2.2, corncob_0.4.1, DESeq2_1.40.2, distinct_1.16.0, edgeR_3.42.4, fastANCOM_0.0.4, glmmTMB_1.1.9, GUniFrac_1.8, limma_3.56.2, Maaslin2_1.14.1, maaslin3_0.99.3, metagenomeSeq_1.42.0, microbiomeMarker_1.6.0, phyloseq_1.46.0, rpart_4.1.23, vegan_2.6-8, ZINQ_2.0

R package for calculating data characteristics: amap_0.8-19, BimodalIndex_1.1.9,

R package for analyzing the performance of DA methods: lme4_1.1-35.1, lmerTest_3.1-3, pROC_1.18.5

**Details on the implementation of the studied methods.** For a brief summary of the 22 evaluated DA tests, we refer to Nearing et al. [1]. Moreover, implementation details are provided in checklist item 18.

**Extent to which code has already been written at the time of pre-registration.** Code to simulate data with metaSPARSim and sparseDOSSA2, calculate data characteristics and apply statistical tests are re-used from a former benchmark study entitled "Leveraging Synthetic Data to Validate a Benchmark Study for Differential Abundance Tests for 16S Microbiome Sequencing Data" which was partly conducted (80%) at the time when this protocol has been written [2]. In that study, we simulated data by calibrating the simulation parameter to each group separately. Since than all taxa have different parameters, there was not meaningful information about whether taxa are regulated and, thus, calculation of sensitivity and specificity were not feasible. All DA methods were tested on at least one experimental dataset, e.g., to provide details about how to call them and how to choose configuration parameters.

### Hardware

We are planning to conduct the analyses on a compute server running with Debian GNU/Linux 12 with a Linux 6.1.0-22-amd64 kernel, equipped with dual AMD EPYC 9454 48-Core Processors (totaling 96 CPUs) and 503 GiB of RAM.

### Reproducibility

**Description of the accessibility and availability of datasets, code, and material after the completion of the study.** Source code of the presented benchmark study and for reproducing the outcomes will be comprehensively published.

**Other effort to ensure or improve reproducibility.** The code development will be performed using git as version control systems. Commits will be performed on a daily basis during the code development phase.

## Prior knowledge, neutrality and dissemination

### Prior knowledge

**Known prior work based on the selected datasets, the analyzed measures in that work and its relation to the planned study.** The major findings from Nearing et al. [1] are:
Different differential abundance (DA) testing methods produce drastically different results when applied to the same microbiome datasets. The number and set of significant amplicon sequence variants (ASVs) identified varied widely across tools. The results also depend on data pre-processing, particularly whether rare taxa are filtered out before analysis. For many tools, the number of features identified correlates with dataset characteristics such as sample size, sequencing depth, and effect size of community differences. Some methods consistently identified more significant features than others, e.g., Limma voom, Wilcoxon (CLR), LEfSe, and edgeR tended to find the largest number of significant ASVs. Compositionally aware methods like ALDEx2 and ANCOM-II were generally found as more conservative, identifying fewer significant ASVs across datasets. ALDEx2 and ANCOM-II tended to identify significant features that were also found by other methods. Their findings highlight the importance of carefully considering methodological choices when conducting and interpreting microbiome differential abundance analyses and underscores the variability in results obtained from different DA testing methods.

Our ongoing benchmark study [2] indicates that metaSPARSim [3] and sparseDOSSA2 [4] can be used to generate datasets with similar data characteristics when all characteristics are considered on an aggregated level (e.g., by using PCA) (Figs 1A and 1B), whereas metaSPARSim [3] generates more similar results compared to sparseDOSSA2 [4]. Moreover, also individual data characteristics mostly coincide for metaSPARSim [3], except the following ones (preliminary results): sparsity, bimodality of the correlation of samples and effect size (Fig 1C). SparseDOSSA2 on the other hand tends to overestimate the library size (Fig 1D). We also see for both simulation tools that adding an appropriate proportion of zeros to the simulated data makes the synthetic data more realistic, i.e., reduces the number of non-equivalent characteristics. In our preliminary results, we also see that the ANCOM-II implementation in the R-package leads to different outcomes as the code used in Nearing et al. [1]. We also started to check, whether conclusions of Nearing et al. [1] can be validates with simulated data. Our preliminary results show that overall trends are reproducible, however only a few conclusions can be validated at a stringent level.

**Prior knowledge about the datasets themselves.** N/A.

### Neutrality statement

**Neutrality statement regarding the investigated methods.** We confirm that all contributors to this study were not involved in the development of the evaluated DA methods, were not involved in developing the two simulation tools and not involved in the experimental studies where the data has been taken from. We also declare that there are no other competing interests that might lead to biased interpretations.

**Steps taken to enhance and/or ensure the neutrality of the study (blinding).** N/A.

### Dissemination

**Plan for publishing the study and its results.** Public access of the generated data, analysis scripts, results and supplemental information is granted as indicated above. The results will be published in a peer-reviewed scientific journal, preferably in the same journal as this protocol.

 

**Availability of results data after the study's completion.** The 38 experimental datasets were downloaded from https://figshare.com/articles/dataset/16S_rRNA_Microbiome_Datasets/14531724 on February 9, 2024. There, Nearing et al. [1] made the datasets from their study available, therefore we incorporate the exact same datasets. We keep a local copy of this data in our Fredato research data management system https://nxc-fredato.imbi.uni-freiburg.de until 31.12.2030 and make it available if the original data is not available in the current version anymore and if this does not violate legal, data protection, or copyright regulations. Generated data, analysis scripts, results and supplemental information to this study will also be stored in the Fredato research data management. In case of unexpected technical limitations, we will make data, analysis scripts and supplemental information available via https://figshare.com.

## Supporting information

**S1 Table: Higher dimension data characteristics.** These are used to calculate the final 46 scalar value characteristics (S2 Table).
(PDF)

**S2 Table: Final integer values data characteristic.**
(PDF)

**S3 Table: Summary information on 38 experimental data templates, which serve as templates for calibrating the simulation tool.**
(PDF)

**S4 Text: Study Protocol Version 1.**
(DOCX)

## Acknowledgments

We would like to thank Prof. Anne-Laure Boulesteix and Julian Lange for our valuable discussions on methodological aspects of benchmark studies and for developing the checklist for methodological studies used for this protocol.

This protocol follows the "Study protocol checklist for real-data methodological research" (v0.9) that has been suggested for documenting methodological research and for reducing bias in benchmark studies [27]. Accordingly, the abstract, introduction, summary of the research question and hypotheses, methods, analysis plans and other relevant sections are provided in the order and with the enumeration of items as proposed in that checklist.

We acknowledge support by the Open Access Publication Fund of the University of Freiburg.

## Author contributions

**Writing – original draft:** Eva Kohnert.

**Writing – review & editing:** Clemens Kreutz.

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
