## [Decision Letter · Decision Letter 0]

3 Jan 2025

PONE-D-24-40619Benchmarking Differential Abundance Tests for 16S Microbiome Sequencing Data Using Simulated Data Based on Experimental TemplatesPLOS ONE

Dear Dr. Kohnert,

Thank you for submitting your manuscript to PLOS ONE. After careful consideration, we feel that it has merit but does not fully meet PLOS ONE’s publication criteria as it currently stands. Therefore, we invite you to submit a revised version of the manuscript that addresses the points raised during the review process.

We look forward to receiving your revised manuscript.

Kind regards,

Stephen D. Ginsberg, Ph.D.

Section Editor

PLOS ONE

2. In your cover letter, please confirm that the research you have described in your manuscript, including participant recruitment, data collection, modification, or processing, has not started and will not start until after your paper has been accepted to the journal (assuming data need to be collected or participants recruited specifically for your study). In order to proceed with your submission, you must provide confirmation.

4. Please ensure that you refer to Figure 3 in your text as, if accepted, production will need this reference to link the reader to the figure.

5. We note you have included a table to which you do not refer in the text of your manuscript. Please ensure that you refer to Table 3 in your text; if accepted, production will need this reference to link the reader to the Table.

Additional Editor comments:

After careful consideration by 2 Reviewers and an Academic Editor, all of the critiques of the Reviewers must be addressed in detail in a revision to determine publication status. If you are prepared to undertake the work required, I would be pleased to reconsider my decision, but revision of the original submission without directly addressing the critiques of the Reviewers does not guarantee acceptance for publication in PLOS ONE. If the authors do not feel that the queries can be addressed, please consider submitting to another publication medium. A revised submission will be sent out for re-review. The authors are urged to have the manuscript given a hard copyedit for syntax and grammar.

**Comments to the Author**

1. Does the manuscript provide a valid rationale for the proposed study, with clearly identified and justified research questions?

Reviewer #1: Yes

Reviewer #2: Yes

2. Is the protocol technically sound and planned in a manner that will lead to a meaningful outcome and allow testing the stated hypotheses?

Reviewer #1: Yes

Reviewer #2: Partly

3. Is the methodology feasible and described in sufficient detail to allow the work to be replicable?

Reviewer #1: Yes

Reviewer #2: Yes

4. Have the authors described where all data underlying the findings will be made available when the study is complete?

Reviewer #1: Yes

Reviewer #2: Yes

5. Is the manuscript presented in an intelligible fashion and written in standard English?

Reviewer #1: Yes

Reviewer #2: Yes

6. Review Comments to the Author

You may also provide optional suggestions and comments to authors that they might find helpful in planning their study.

Reviewer #1: The author proposes a study to benchmark various differential abundance (DA) analysis methods for microbiome data. It has two parts : (1) simulation framework by incorporating a known truth into the synthetic dataset, (2) report the accuracy of various DA methods to recover the ground truth, assess the performance of DA methods and contributors with varying data characteristics like sparsity, effect size, sample size.

1. What is the rationale for simulating data using these two methods metaSPARsim and sparseDOSSA compared to other frameworks available like MIDASim, DM. It would be great if you could more about this methods in the paper.

2. Author plans to test the Aim 1 hypothesis by using the synthetic data with known ground truth. I am interested to know if it is possible to add in few confounders, no effect cases or known negative cases as well to evaluate FDR of these methods.

3. If the above is not feasible, please comment on how you plan to comment on FDR of these methods.

4. I am interested to see the results of Aim3. Potentially if we can setup some rule based selection criteria to recommend a DA method based on data characteristics.

Reviewer #2: This proposed protocol by Kohnert and Kreutz proposes to validate and benchmark differential abundance (DA) methods based on simulated data produced by sparseDOSSA2 and metaSPARSim using 38 datasets from Nearing et al. 2022. In a prior protocol report, the authors proposed a methodology for validating simulated datasets to ensure their resemblance to real-world data. This work looks to expand on Nearing et al.’s report and their previous protocol proposal by investigating the incongruence between different DA tools on simulated data with known ground truths on the same 16S rRNA gene sequencing datasets used in Nearing et al., 2022. They have three main aims in their study which in general are to 1) investigate the performance of DA methods using known ground truths in simulated data, 2) to identify whether effect size, sparsity, and sample size have impact on DA performance and 3) to identify dataset characteristics that may be associated with DA method performance. This timely protocol submission will interest the microbiome community and could help future users identify the appropriate DA method for their datasets.

Major comments:

The authors claim good agreement between the real and simulated datasets in lines 172-174, but the reference provided is merely a protocol without actual results. Including preliminary results and or some other reference here would be beneficial, as part of the study's conclusions relies on the resemblance between real-world and simulated data.

The explanation of methods to generate known ground truths for each simulated dataset is unclear and would benefit from more detail, possibly with a figure. Step (2) at line 226 poses a risk of bias, as selecting a random p-value from a family of related tools (like the two limma-voom methods) might skew results toward that family. The authors should heavily consider this aspect of their design. Additionally, comparing results from a NULL dataset with those from one with a known ground truth could be informative (i.e. if sparssDOSSA2 is used to make a null distribution from template X do we expect any tools to identify significant features).

The authors plan to use the same DA methods originally published in Nearing et al., 2022. Although many of these methods remain popular, there have been significant updates and new tools introduced since then. To enhance the report's relevance, the authors might consider including newer tools like LOCOM, MaAsLin3, or ANCOM-BCII.

Minor comments:

Lines 346-352: I would like if the authors could give more details on why they choose this normalization and why it is appropriate for use with metaSPARSim.

It is unclear why the authors choose to alter the code for LEfSe testing given most users of this tool would not go through the lengths required to generate p-values for all features. Given this it may be more appropriate to use default settings to reflect real world use (and forgive fdr correction as was done in Nearing et al., 2022).

Lines 567-570: It is unclear how the dataset splitting will occur and how we expect this to effect the resulting data. Please give more details about this.

Lines 658-667: Indicates results that are ongoing but cannot be assessed and must be taken at face value. It makes evaluating this section difficult. Some preliminary data to justify these results would be helpful.

7. PLOS authors have the option to publish the peer review history of their article (what does this mean? ). If published, this will include your full peer review and any attached files.

**Do you want your identity to be public for this peer review?**  For information about this choice, including consent withdrawal, please see our Privacy Policy .

Reviewer #1: No

Reviewer #2: No

---

## [Author Response · Author response to Decision Letter 1]

17 Feb 2025

Responses to Reviewers:

Reviewer #1: The author proposes a study to benchmark various differential abundance (DA) analysis methods for microbiome data. It has two parts : (1) simulation framework by incorporating a known truth into the synthetic dataset, (2) report the accuracy of various DA methods to recover the ground truth, assess the performance of DA methods and contributors with varying data characteristics like sparsity, effect size, sample size.

Comment reviewer:

1. What is the rationale for simulating data using these two methods metaSPARsim and sparseDOSSA compared to other frameworks available like MIDASim, DM. It would be great if you could more about this methods in the paper.

Response from authors:

When planning the study and preparing the protocol, we screened the literature for availability and feasibility of appropriate simulation tools. The requirements were

1) possibility for calibrating simulation parameters using real datasets as template

2) generation of count data

3) convincing methodology

4) availability in R or possibility to integrate it in R (e.g. via interface or command line)

Altogether, we found and evaluated 11 simulation methods and identified metaSPARSim and spareseDOSSA2 as mostly suitable.

MIDASim was not included in our evaluation because it was published later.

Based on the reviewer’s suggestion, we now decided to also include MIDASim. However, due to limitations in terms of computational efforts, we will select the two of these tools that most realistically generate simulated data, i.e. with the least number of non-equivalent data characteristics.

For simulating count matrices according to the Dirchilet-Multinomial (DM) models, we could not find a package that enables calibration of the simulation parameters based on a given experimental count matrix.

Comment from reviewer:

2. Author plans to test the Aim 1 hypothesis by using the synthetic data with known ground truth. I am interested to know if it is possible to add in few confounders, no effect cases or known negative cases as well to evaluate FDR of these methods.

Response from author:

The reviewer hints to two important generalizations, namely more complex study designs comprising multiple covariates (such as confounders), and no effect or negative cases that could be seen as samples assigned to the wrong group.

1) Concerning multivariate analyses: We defined the scope of our benchmark study to two-group designs (=one covariate) as in Nearing et al. and our previous/ongoing benchmark study. Extending the scope of this study to multivariate settings would require knowledge about further covariates and would require further assumptions about the effect size distributions introduced by these covariates and about the proportion of samples and taxa affected by covariates/confounders. Moreover, the multi-colinearity of all covariates have to be chosen realistically. For these assumptions, there is no knowledge available, but the choice can have a noticeable impact on our benchmarking outcomes. So including such aspects would strongly increase the complexity of the study and might leave challenges that can be hardly solved.

A very general recommendation of rigorous planning of research studies is to focus on a primary hypothesis and not address too many questions jointly. In line with this general guideline, we concentrate all our efforts on two-group comparisons.

Another aspect is that, simulating data for multiple covariates is not supported by most (or even all) simulation tools. However, the tools intend to generate realistic data for each individual sample in a dataset, so they intend to consider the effects of all covariates together on each sample, but do not “decompose” (estimate) individual effects which would be necessary for controlling the underlying true effects. So, overall the distribution of the data is presumed to reflect also effects of hidden covariates.

2) Concerning known negative cases: Since we use real experimental dataset for calibrating the parameters of the simulation tools, the synthetic data that is simulated effectively contains the same proportion of negative cases (samples) as they occur in the experimental template. Taxa with zero effects between both groups of samples are also considered by simulating the calibrated proportion of non-differentially abundant taxa. This aspect is described in more detail in the new protocol version.

Comment from reviewer:

3. If the above is not feasible, please comment on how you plan to comment on FDR of these methods.

Response from author:

FDR is evaluated by looking at non-differentially abundant taxa.

As pointed out above, the new version of our protocol includes an improved explanation of how we simulate appropriate proportions of differentially and non-differentially abundant taxa.

Comment from reviewer:

4. I am interested to see the results of Aim3. Potentially if we can setup some rule based selection criteria to recommend a DA method based on data characteristics.

Response from author:

In line with the reviewer’s suggestion, we planned to derive decision trees. In the new protocol version, we provide a more detailed description and explanation.

Reviewer #2: This proposed protocol by Kohnert and Kreutz proposes to validate and benchmark differential abundance (DA) methods based on simulated data produced by sparseDOSSA2 and metaSPARSim using 38 datasets from Nearing et al. 2022. In a prior protocol report, the authors proposed a methodology for validating simulated datasets to ensure their resemblance to real-world data. This work looks to expand on Nearing et al.’s report and their previous protocol proposal by investigating the incongruence between different DA tools on simulated data with known ground truths on the same 16S rRNA gene sequencing datasets used in Nearing et al., 2022. They have three main aims in their study which in general are to 1) investigate the performance of DA methods using known ground truths in simulated data, 2) to identify whether effect size, sparsity, and sample size have impact on DA performance and 3) to identify dataset characteristics that may be associated with DA method performance. This timely protocol submission will interest the microbiome community and could help future users identify the appropriate DA method for their datasets.

Author:

We thank the reviewer for this positive and appreciative feedback.

Major comments:

Comment from reviewer:

The authors claim good agreement between the real and simulated datasets in lines 172-174, but the reference provided is merely a protocol without actual results. Including preliminary results and or some other reference here would be beneficial, as part of the study's conclusions relies on the resemblance between real-world and simulated data.

Response from author:

According to the reviewer’s suggestion, we included a figure with several plots summarizing our preliminary findings into our protocol.

Comment from reviewer:

The explanation of methods to generate known ground truths for each simulated dataset is unclear and would benefit from more detail, possibly with a figure.

Response from author:

In the new version of our protocol, we now provide a more detailed explanation.

Comment from reviewer:

Step (2) at line 226 poses a risk of bias, as selecting a random p-value from a family of related tools (like the two limma-voom methods) might skew results toward that family. The authors should heavily consider this aspect of their design.

Response from author:

We agree with the reviewer, that this constitutes a risk of bias. However, any other approach would lead to risk of bias. As an example, taking the same number of p-values for each family of tests, would penalize tests that are in such families or in larger families. Moreover, multiple tests are to different extend based on similar assumptions. Thus defining „families“ is hardly feasible.

Because of two reasons, we think that the risk of bias in the suggested approach is small: 1) The number of tests has been increases from 14 to 22. Within the 22 tests, we only consider a maxium of two tests that are strongly related (e.g. the two limma voom approaches). In case the anticipated bias exists, the impact should be small.

2) Importantly, the tests at this analysis stage only enter in the decision about how many taxa will be simulated as different and in the assignment of the taxa to be simulated as differential or non-differential. Data generation including the noise of the simulated count data is then independent of these choices.

We hope that the reviewer's concerns are resolved with this explanation and the improved description of the planned simulation procedure in the protocol.

Comment from reviewer:

Additionally, comparing results from a NULL dataset with those from one with a known ground truth could be informative (i.e. if sparssDOSSA2 is used to make a null distribution from template X do we expect any tools to identify significant features).

Response from author:

This concern might also partly originate from our insufficient explanation of the simulation procedure: Each simulated data set will have a proportion (1-pi0) of differentially abundant taxa, and a proportion pi0 of *not* differentially abundant taxa that are generated from the null distribution obtained by calibrating the simulation tool to all samples. This subset of taxa will be used to evaluate the FDR.

A dataset consisting of solely not differentially abundant taxa could in principle also be used to evaluate the desired FDR levels, but does not allow for assessing the TPR or evaluating the trade-off between sensitivity and specificity, for example in terms of the pAUC.

Comment from reviewer:

The authors plan to use the same DA methods originally published in Nearing et al., 2022. Although many of these methods remain popular, there have been significant updates and new tools introduced since then. To enhance the report's relevance, the authors might consider including newer tools like LOCOM, MaAsLin3, or ANCOM-BCII.

Response from author:

Yes, we agree with the reviewer and decided to extend the study to all DA methods that are currently available as a package in R. We found and now include also the following tools: ADAPT, ANCOM-BC2, distinctTest, fastANCOM, glmmTMB, linDA, , MaAsLin3, ZicoSeq, and ZINQ. Altogether, we now plan to evaluate 22 DA methods instead of 14.

We did not include LOCOM because it failed to be installed in its latest version. We also did not include MECAF since it is not yet available as an R package and we did not include combinations of two methods such as ZINB-WaVE + DESEQ2 that have to be combined “manually” by several lines of own code. We also did not include methods that were replaced by a more advanced method such as ANCOM-BC that was improved by an improved bias correction and zero-inflation modelling in ANCOM-BC2.

Minor comments:

Comment from reviewer:

Lines 346-352: I would like if the authors could give more details on why they choose this normalization and why it is appropriate for use with metaSPARSim.

Response from author:

We now provide a more detailed explanation:

In short, calibration of the simulation tools was conducted as suggested in their manuals. Accordingly, we use GMPR for the calibration of metaSPARSim, as it requires both raw and normalized data. GMPR is also the normalization approach employed in the metaSPARSim vignette.

For testing differential expression, each DA method was applied with the default normalization approach recommended for the respective tool.

Comment from reviewer:

It is unclear why the authors choose to alter the code for LEfSe testing given most users of this tool would not go through the lengths required to generate p-values for all features. Given this it may be more appropriate to use default settings to reflect real world use (and forgive fdr correction as was done in Nearing et al., 2022).

Response from author:

We proposed the described procedure to handle the fact that on the one hand, we need all p-values to calculate pAUC but LEfSe does not output p-values for taxa that do not satisfy the default cutoff values. On the other hand, we discovered that LEfSe can fail if those cutoffs deviate from the default. The strategy we proposed so far, was to relax those cutoffs and increase it iteratively until LEfSe succeeds. Nearing et al. also had to deal with this issue and chose the following strategy: "From these, only those features with scaled LDA analysis scores above the threshold score of 2.0 (default) were called as differentially abundant. This key step is what distinguished LEfSe from the Wilcoxon test approach based on relative abundances that we also ran. In addition, no multiple-test correction was performed on the raw LEfSe output as only the p-values of significant features above-threshold LDA scores are returned by this tool."

Another aspect that should be considered is, that FDR should not be computed after any kind of filtering that is related to significance of the DA methods.

Despite these arguments, we agree with the reviewer that we should apply LEfSe to reflect real world use as good as possible. We therefore decided to run LEfSe with relaxed cutoffs to obtain all p-values. However, if LEfSe fails for >10% of the data sets, but succeeds with default cutoffs, we use this outcome instead. In any case, we calculate FDR from to those (possibly filtered) p-values, independently on violation of mathematical assumptions since this would presumably also be done in real-world applications. We changed the protocol accordingly.

Comment from reviewer:

Lines 567-570: It is unclear how the dataset splitting will occur and how we expect this to effect the resulting data. Please give more details about this.

Response from author:

We apologize for this and have now provided a clearer and more detailed description in the revised version of our protocol.

Comment from reviewer:

Lines 658-667: Indicates results that are ongoing but cannot be assessed and must be taken at face value. It makes evaluating this section difficult. Some preliminary data to justify these results would be helpful.

Response from author:

In order to justify that metaSPARSim and sparseDOSSA can be used to generate data with similar data characteristics, we now provide boxplots of the most important individual data characteristics and PCA calculated from all data characteristics.

---

## [Decision Letter · Decision Letter 1]

7 Mar 2025

Benchmarking Differential Abundance Tests for 16S Microbiome Sequencing Data Using Simulated Data Based on Experimental Templates

PONE-D-24-40619R1

Dear Dr. Kohnert,

We’re pleased to inform you that your manuscript has been judged scientifically suitable for publication and will be formally accepted for publication once it meets all outstanding technical requirements.

Kind regards,

Stephen D. Ginsberg, Ph.D.

Section Editor

PLOS ONE

**Comments to the Author**

1. Does the manuscript provide a valid rationale for the proposed study, with clearly identified and justified research questions?

Reviewer #1: Yes

2. Is the protocol technically sound and planned in a manner that will lead to a meaningful outcome and allow testing the stated hypotheses?

Reviewer #1: Yes

3. Is the methodology feasible and described in sufficient detail to allow the work to be replicable?

Reviewer #1: Yes

4. Have the authors described where all data underlying the findings will be made available when the study is complete?

Reviewer #1: Yes

5. Is the manuscript presented in an intelligible fashion and written in standard English?

Reviewer #1: Yes

6. Review Comments to the Author

You may also provide optional suggestions and comments to authors that they might find helpful in planning their study.

Reviewer #1: Thanks for addressing and answering questions from the previous round of reviews. I appreciate your efforts in preparing the manuscript and carrying out the work to help microbiome research identify the appropriate DA methods.

1. Thank you for including the newer MIDASim method and expanding the study to cover more DA methods in the revised manuscript.

2. Thanks for including the methods planned for AIM3. Hopefully, this will guide the microbiome researchers in selecting appropriate methods for analysis.

7. PLOS authors have the option to publish the peer review history of their article (what does this mean? ). If published, this will include your full peer review and any attached files.

**Do you want your identity to be public for this peer review?**  For information about this choice, including consent withdrawal, please see our Privacy Policy .

Reviewer #1: No

---

## [Editor Report · Acceptance letter]

PONE-D-24-40619R1

PLOS ONE

Dear Dr. Kohnert,

I'm pleased to inform you that your manuscript has been deemed suitable for publication in PLOS ONE. Congratulations! Your manuscript is now being handed over to our production team.

Kind regards,

on behalf of

Dr. Stephen D. Ginsberg

Section Editor

PLOS ONE